# Role of Doping Agent Degree of Sulfonation and Casting Solvent on the Electrical Conductivity and Morphology of PEDOT:SPAES Thin Films

**DOI:** 10.3390/polym13040658

**Published:** 2021-02-23

**Authors:** Daniela Valeria Tomasino, Mario Wolf, Hermes Farina, Gianluca Chiarello, Armin Feldhoff, Marco Aldo Ortenzi, Valentina Sabatini

**Affiliations:** 1Department of Chemistry, Università degli Studi di Milano, Via Golgi 19, 20133 Milano, Italy; daniela.v.tomasino@gmail.com (D.V.T.); hermes.farina@unimi.it (H.F.); gianluca.chiarello@unimi.it (G.C.); marco.ortenzi@unimi.it (M.A.O.); 2Institute of Physical Chemistry and Electrochemistry, Leibniz University Hannover, Callinstrasse 3A, 30167 Hannover, Germany; wolf.mario@gmx.de (M.W.); armin.feldhoff@pci.uni-hannover.de (A.F.); 3Consorzio Interuniversitario per la Scienza e Tecnologia dei Materiali (INSTM), Via Giusti 9, 50121 Firenze, Italy; 4CRC Materiali Polimerici “LaMPo”, Dipartimento di Chimica, Università degli Studi di Milano, Via Golgi 19, 20133 Milano, Italy

**Keywords:** sulfonated polyarylether sulfone, poly(3,4-ethylenedioxythiophene), doping agent, degree of sulfonation, casting solvent effect, electrically conductive thin films

## Abstract

Poly(3,4-ethylenedioxythiophene) (PEDOT) plays a key role in the field of electrically conducting materials, despite its poor solubility and processability. Various molecules and polymers carrying sulfonic groups can be used to enhance PEDOT’s electrical conductivity. Among all, sulfonated polyarylether sulfone (SPAES), prepared via homogenous synthesis with controlled degree of sulfonation (DS), is a very promising PEDOT doping agent. In this work, PEDOT was synthesized via high-concentration solvent-based emulsion polymerization using 1% *w*/*w* of SPAES with different DS as dopant. It was found that the PEDOT:SPAESs obtained have improved solubility in the chosen reaction solvents, i.e., *N*, *N*-dimethylformamide, dimethylacetamide, dimethyl sulfoxide, and *N*-methyl-2-pyrrolidone and, for the first time, the role of doping agent, DS and polymerization solvents were investigated analyzing the electrical properties of SPAESs and PEDOT:SPAES samples and studying the different morphology of PEDOT-based thin films. High DS of SPAES, i.e., 2.4 meq R-SO_3_^−^
× g^−1^ of polymer, proved crucial in enhancing PEDOT’s electrical conductivity. Furthermore, the DMSO capability to favor PEDOT and SPAES chains rearrangement and interaction results in the formation of a polymer film with more homogenous morphology and higher conductivity than the ones prepared from DMAc, DMF, and NMP.

## 1. Introduction 

In the field of electrically conducting polymers (CPs), in the last decade, poly(3,4-ethylenedioxythiophene) (PEDOT) took the leading role for emerging technologies, such as organic electronics, printed circuits, and flexible electronics [1]. It is a *p*-type semiconductor [2] that can be obtained by the oxidative polymerization of 3,4-ethylenedioxythiophene (EDOT) following either an electrochemical or a chemical route. PEDOT polymerization is also performed in the presence of an agent carrying sulfonic moieties used both as a stabilizer and a dopant for PEDOT polymers, with a doping concentration variable from 1% to 10% *w*/*w*. The dopants based on sulfonic groups commonly used with PEDOT are either organic molecules such as 2-naphthalene sulfonic acid and para toluene sulfonic acid or sulfonated polymers, i.e., sulfonated polystyrene (PSS) [3]. PSS is currently the most successful PEDOT dopant, since works as a template and dopant in PEDOT: PSS dispersion, and provides water solubility to PEDOT. However, insulating properties of PSS pose several issues when it is used as the template: in fact, this makes it difficult to balance the conductivity and processing ability of PEDOT: PSS dispersion. A high PSS-to-PEDOT ratio results in good stability of the dispersion and film formation, but decreases conductivity of the resulting film [3]. It is desirable to harness the useful properties of PEDOT and its derivatives in conductive patterns and thus there is a need to discover new types of PEDOT doping agents to solve the above-mentioned problems.

However, doped PEDOT films have been the focus of research for several electronic and thermoelectric purposes in recent years [4,5,6]. PEDOT’s major drawbacks are related to its insolubility in all common inorganic and organic solvents and, as all CPs, to its low electrical conductivity when in the undoped state [1,7]. To change PEDOT oxidation state, and hence to optimize its electric conductivity features, several studies focused on its modification by performing doping and de-doping treatments [8,9]. Different values of the electrical conductivity can be achieved either by tuning the doping agent ratio and/or by introducing secondary dopants, i.e., high boiling point solvents, ionic liquids, and surfactants [10,11]. Additionally, relatively few works have been reported on PEDOT films forming techniques [12,13,14], particularly concerning the relationship between the doping agent chosen and the role of the casting solvent onto PEDOT-based thin films resulting properties.

In this scenario, also ionomeric polymers are becoming key materials for the preparation of membranes to be used in a wide variety of applications such as water purification, transducers, and energy conversion technology [15]. To be efficient, ion-exchange membranes need to have high permselectivity, low electrical resistance, good physicochemical stability, and satisfactory mechanical properties, the latter commonly obtained using crosslinked structures [16,17]. Among all the commercially available ionomers, Nafion^®^ is nowadays the most widely adopted. It consists of a sulfonated tetrafluoroethylene-based fluoropolymer, mostly used as a protonic exchange membrane (PEM) for fuel cell applications. The presence of R-SO_3_H groups along the hydrophobic polytetrafluoroethylene backbone confers both proton conductivity and hydrophilicity features to the material. It is consequently chemically inert in both oxidative and reductive environments and its hydrophilicity–hydrophobicity balance guarantees good stability in the presence of water and elastic strength [18,19]. Nevertheless, it presents many limitations such as high cost of fabrication, limited operating temperature range (around 80 °C), poor barrier properties towards methanol, and a high osmotic drag [19,20,21]. Thus, there is still a need to develop new types of ionomers that could overcome these issues.

The electrical conductivity of a PEM depends not only on the host polymer structure but also on its water content. In this respect, polyarylether sulfone (PAES)-based membranes, which are widely used in several industrial and biomedical separation processes [22,23,24,25], are gaining more and more interest thanks to the improved wetting properties achieved by incorporating a controlled amount of sulfonic groups along PAES polymer chains [26]. Sulfonated PAES (SPAES) results being a thermoplastic polymer with excellent thermal and mechanical properties, photochemical stability, and easy processability [24,27] and can therefore be effectively used in PEMs. It can be obtained by a heterogeneous approach of post-sulfonation modification of PAES or via a homogeneous synthesis consisting of a one-pot copolymerization reaction in the presence of a sulfonated co-monomer [20,21,26,28,29]. In our previous works [26,30], it was demonstrated that SPAES with controlled degree of sulfonation (DS) can be obtained via homogeneous synthesis by condensation reaction in the presence of 2,5-dihydroxybenzene-1-sulfonate potassium salt as sulfonated comonomer. In such a way, a direct correlation between the DS, the thermal properties, and the water content of the final material was demonstrated, leading to a tunable final hydrophilicity of polymers with a simultaneously high thermal stability [26,30].

Even though SPAES is nowadays commercially used in a variety of membrane processes, e.g., ion exchange membranes, reverse osmosis, and electrodialysis processes [22], the previously mentioned properties make it an interesting material for electroanalytic and electroconductive purposes. For example, Falciola et al. compared SPAES membranes to Nafion^®^ as electrode modifiers for the detection of heavy metals, such as lead and ruthenium, in water [31]. Furthermore, Sabatini et al. described the use of SPAES-based inks for the preparation of printed, flexible, and electrically conductive thin films [32,33].

In our previous works [34,35], it was reported that SPAES can be successfully used as a PEDOT doping agent. Here, for the first time, a correlation between the SPAES DS and the electrical conductivity of PEDOT: SPAES thin films obtained via spin-coating using different polar aprotic solvents is established.

SPAESs with increasing DS (from 0.75 to 2.4 meq R-SO_3_^−^
× g^−1^ of polymer) were synthesized via direct condensation copolymerization, and their structure and real DS were determined via ^1^H NMR spectroscopy.

SPAESs ion exchange capacity was measured by potentiometric titration and the thermal properties were investigated by differential scanning calorimetry analyses. SPAES dielectric properties were studied by impedance spectroscopy, determining the direct relationship between SPAES electrical conductivity and DS.

As reference, SPAESs with the lower and higher DS were used as dopants for PEDOT (at 1% *w*/*w* of SPAES with respect to EDOT) to determine, maintaining SPAES concentration unaltered, the influence of different amounts of sulfonic groups onto the resulting electrical conductivity. Lastly, to better understand the influence of the solvents dipolar moment and chemical structure over the polymer chains rearrangement, hence over the final electrical conductivity of PEDOT foils, a morphological evaluation of PEDOT: SPAES_2.4 thin films prepared in DMF, DMAc, DMSO, and NMP as casting solvents was performed via scanning electron microscopy (SEM) analysis.

## 2. Experimental

### 2.1. Materials

3,4-Ethylenedioxythiophene (EDOT, >97%), ferric sulfate (Fe_2_(SO_4_)_3_·xH_2_O, >97%) 4,4’-difluorodiphenylsulfone (BFPS, ≥99%), 4,4’-dihydroxydiphenyl (BHP, ≥97%), benzoic acid (≥99.5%), sodium chloride (NaCl, >99%) were supplied by Sigma Aldrich. 1,2,3-trihydroxybenzene (THB, 98%) and potassium carbonate (K_2_CO_3_, ≥98%) were purchased by Fluka; 2,5-dihydroxybenzene-1-sulfonate potassium salt (sulfonated hydroquinone, SHQ, ≥98%) was obtained from Alfa Aesar (Thermo Fisher GmbH, Kandel, Germany). All the reagents were dried at 30 °C in vacuum oven (about 4 mbar) for at least 24 h before the use and employed without further purification. *N*,*N*-dimethylformamide (DMF, 99.8% anhydrous), dimethylacetamide (DMAc, ≥99.5%), dimethyl sulfoxide (DMSO, ≥99.5%), *N*-methyl-2-pyrrolidone (NMP, ≥99.5% anhydrous), toluene (99.8% anhydrous), distilled water Chromasolv^®^ (≥99.9%), hydrochloric acid (1.00 M HCl), standard sodium hydroxide solution (0.01 M NaOH), and dimethyl sulfoxide-*d_6_* (DMSO-*d_6_*, 99.96 atom % D) were supplied by Sigma Aldrich (Merck Life Sciences, Milan, Italy) and used without further purification.

### 2.2. Synthesis of Sulfonated Polyarylether Sulfone (SPAES)

SPAESs having a nominal DS of 0.75-1.0-1.3-1.6-1.9-2.1-2.4 meq R-SO_3_^−^
× g^−1^ of polymer (named SPAES_0.75, 1.0, 1.3, 1.6, 1.9, 2.1, and 2.4, respectively), were synthesized as reported in our previous works [26,30]. The exact amounts of monomers used for the syntheses are reported in Table 1. Briefly, in a representative polymerization procedure, BPFS, BHP, SHQ, THB in the case of SPAES_1.3, 1.6, 1.9, 2.1, and 2.4 and K_2_CO_3_, the latter used as proton scavenger, were introduced into a 100 cm^3^ one neck round bottom flask equipped with a modified Dean-Stark device and a magnetic stirring bar, under nitrogen atmosphere.

The flask, loaded with toluene and NMP in order to have a 10% *w*/*V* concentration of the monomers in the solution, was put in an oil bath and the polymerization reaction was carried out under reflux for 6 h; the water formed during the reaction was removed as an azeotrope with toluene through the modified Dean-Stark device. After the complete removal of water, the temperature was increased until 198 °C with a step of 10 °C every 30 min and then the reaction mixture was kept at 198 °C for 18 h. A hot viscous, dark-purple solution was obtained and precipitated into a large excess of cold water. The brown solid obtained was recovered via filtration and washed with cold water to remove residual monomers, solvents, and K_2_CO_3_ traces. Due to the great difficulty in removing solvents like NMP and toluene from SPAESs, the samples were washed with cold water for a week and then dried in a vacuum oven (around 4 mbar) at 200 °C for 24 h. The presence of residual solvents was checked via isothermal thermogravimetric analysis (TGA) under nitrogen flow for 2 h at 200 °C. SPAESs was converted to the acid form by immersion of each sample in 50 cm^3^ of 1M HCl solution for 24 h at 100 °C, followed by washing with water for 24 h at 25 °C and drying in a vacuum oven (around 4 mbar) at 150 °C for 24 h.

### 2.3. Synthesis of PEDOT Doped with SPAES

SPAES_0.75 and SPAES_2.4 were used as PEDOT doping agents. The reaction between PEDOT and SPAES was performed varying the reaction solvent, i.e., DMF, DMAc, DMSO, or NMP, according to our previous works [31,35]. In brief, a 50 cm^3^ glass test tube equipped with magnetic stirring was loaded with EDOT (2.50 g), Fe_2_(SO_4_)_3_·xH_2_O (5 g) as radical initiator, SPAES (0.025 g) at 1% *w*/*w* with respect to EDOT monomer and the reaction solvent (3 cm^3^). The polymerization mixture was stirred for 24 h at room temperature under air atmosphere. The obtained polymerization products were analyzed as obtained.

### 2.4. Characterization of SPAESs and PEDOT_SPAESs

#### 2.4.1. Nuclear Magnetic Resonance (^1^H-NMR) Spectroscopy

^1^H NMR spectra were collected at 25 °C with a BRUKER 400 MHz spectrometer. Samples for the analyses were prepared dissolving 9–10 mg of polymer in 1 cm^3^ of DMSO-*d_6_*. The presence of R-SO_3_^−^ groups in SPAES samples was quantitatively measured via ^1^H NMR spectroscopy calculating the integral ratios between the proton in ortho to the sulfonic group [g] of SHQ and the ones of BFPS [d], [f], and of BHP [a], as reported in Figure 1 for ^1^H NMR spectrum of SPAES_0.75 and 2.4, using Equation (1), where I_g_: integral area of peak [g]; I_d,f_: integral area of the peaks [d] and [f]; I_a_: integral area of the peak [a]; Ur_BFPS_: molecular weight of BFPS repeat unit −216.25 g × mol^−1^; Ur_BP_: molecular weight of BP repeat unit (184.21 g × mol^−1^); Ur_SHQ_ molecular weight of SHQ repeat unit (226.26 g × mol^−1^):DS = (I_g_ × 1000)/(((I_d,f_ × Ur_BFPS_)/4) + ((I_a_ × Ur_BP_)/4) + (I_g_ × Ur_SHQ_))(1)

#### 2.4.2. Potentiometric Titration (PT)

The presence of R-SO_3_^−^ groups in SPAES samples in terms of ion exchange capacity (IEC) was also determined via potentiometric titration (PT) [36]. A solution of benzoic acid (60 mg in 50 cm^3^ of water) was prepared. A 0.01 M NaOH solution was used for the titration of 75 cm^3^ of water; the NaOH solution volume used for the titration is indicated as *V*_H2O_ (cm^3^); then, 10 cm^3^ of benzoic acid solution were titrated, obtaining the NaOH solution volume used for the neutralization, *V*_benz_ (cm^3^). Solvent factor (*f*) was calculated using Equation (2):f = m/(V_titr_ × M_benz_ × [NaOH]).(2)
where *m* is the weight of benzoic acid in solution (g); V_titr_ is given by the difference between *V*_benz_ and *V*_H2O_ (cm^3^); [NaOH] is the concentration of 0.01 M NaOH solution (mol × cm^−3^); and M_benz_ is the molecular weight of benzoic acid (122.12 g×mol^−1^). SPAESs IEC was measured after soaking the protonated sample in 20 cm^3^ of 2M NaCl solution for at least 24 h. The solution was then titrated with 0.01 M NaOH using a Titrino751 GPD automatic potentiometric titrator (Metrohm). IEC data can be calculated with Equation (3) (where [NaOH]: 0.01 M NaOH solution; V: NaOH volume used during the neutralization of each sample; *f*: solvent factor obtained as described in Equation (2); *x*: weight of sample (g)):IEC = ([NaOH] × V × f)/x.(3)

#### 2.4.3. Differential Scanning Calorimetry (DSC)

Differential scanning calorimetry (DSC) analyses were conducted using a Mettler Toledo DSC 1, on SPAES samples weighing 9–10 mg and under nitrogen atmosphere. According to Checchia et al. [30], the glass transition temperature (*T*_g_) of the samples was measured using the following temperature program: (i) heating from 25 to 300 °C at 10 °C/min; (ii) 5 min isotherm at 300 °C; (iii) cooling from 300 to 25 °C at 10 °C/min; (iv) 5 min isotherm at 25 °C; and (v) heating from 25 to 300 °C at 10 °C/min (*T*_g_ was measured here).

#### 2.4.4. Impedance Spectroscopy

SPAESs pellets were prepared by pressing 45 mg of polymer powder (hydraulic press, pressure: 40 kN). In the case of SPAES_2.4, it was necessary to solubilize the powder in 0.5 cm^3^ DMF. Dielectric measurements were performed at 25 °C. The frequency range employed was 10^7^–10^−2^ Hz and the analyses were performed at constant alternating current (AC) voltage of 1 V.

#### 2.4.5. Laser Scanning Microscopy (LSM)

The thickness of PEDOT_SPAES films obtained in DMSO was determined using a confocal laser scanning microscope (LEICA DCM 3D) with Mirau interferometer. The films were prepared by spin coating technique (12.5 µL, 7000 rpm, 40 s) on SiO_2_ wafer (0.5 cm × 0.5 cm).

#### 2.4.6. Electrical Conductivity

The electrical conductivity of PEDOT_SPAES_0.75 and 2.4 films was measured with a home-made measurement cell in a three-zone horizontal oven with an ELITE thermal system and KEITHLEY 2100 Digital Multimeters. Polymeric films were prepared by spin-coating technique (7000 rpm, 40 s); their thickness is based on the results obtained from the LSM analysis and an average thickness of 25 nm has been assumed. For contacts, commercial silver ink has been used. The temperature dependence of the electrical resistance was measured in the temperature range (313–375 K) and then converted into electrical conductivity values by applying Equation (4):(4)1/σ = R× (A/L).
where *σ* is the electrical conductivity (S × cm^−1^), *R* is the resistance (Ω), *A* is the cross-sectional area (cm^2^), and *L* is the length of the silicon wafer (cm).

#### 2.4.7. Scanning Electron Microscopy (SEM)

Scanning electron microscopy (SEM) micrographs of PEDOT: SPAES films were obtained on a field-emission instrument of the type JEOL JSM-6700F (JEOL, Akishima Tokyo, Japan). Secondary electron imaging of the samples was analyzed using an acceleration voltage of 2 kV and an acceleration current of 10 μA.

## 3. Results and Discussion

### 3.1. SPAESs Synthesis and Characterization: ^1^H-NMR, PT, DSC, and Electrical Conductivity Analyses

Two linear SPAES with a nominal DS of 0.75 and 1.0 meq R-SO_3_^−^
× g^−1^ of polymer (namely SPAES_0.75 and 1.0) and five branched ones with a nominal DS of 1.3, 1.6, 1.9, 2.1, and 2.4 (namely SPAES_1.3, 1.6, 1.9, 2.1, and 2.4) were synthesized by an aromatic nucleophilic substitution reaction. Like other aromatic polymers, due to the combination of high DS and an amorphous phase, SPAES struggles to combine high ionic transport with mechanical stability [30]. In fact, increasing the IEC performances through a higher DS improves the continuity between ionic centers, though at the cost of large water swelling and consequent membrane mechanical weakness. One possible answer to excessive swelling is to stabilize highly hydrophilic polymers by changing their macromolecular architecture [30]. In this contest, we proposed the homogeneous synthesis of branched SPAES characterized by a low concentration of a trifunctional monomer, i.e., THB, conferring the desired properties through the copolymerization reaction. By this approach, we pursued the control of swelling by means of the feed concentrations of a sulfonated monomer and the degree of branching, respectively. Homogeneous synthesis of SPAES is usually performed through an A2 + B2 polycondensation, using di-halogenated (A2) and diol (B2) monomers; to attain high conversion, the homogeneous synthesis of a branched copolymer needs to take into account the reactivity of the B3 branching comonomer and possible side reactions, while not pushing the reaction towards crosslinked products.

Here, a series of branched SPAES copolymers (namely SPAES_1.3, 1.6, 1.9, 2.1, and 2.4) was synthesized using the 0.1% mol/mol of THB with respect to the A2 monomer, i.e., BFPS. The copolymerization balance can thus be written as A2 + B2 + (3y/2) A2 + yB3, where A stands for the aryl halide reactive group, B for the aryl hydroxyl group, and y for concentration of the trifunctional B3 monomer (Figure 1).

The reasons for using THB low concentrations are to effect a minimal change to the course of the polycondensation with respect to the synthesis of linear SPAES and to progressively reduce the swelling of highly sulfonated SPAES while avoiding gelation and thus the presence of insoluble polymers. Through this approach, crosslinking issues were avoided and branched SPAES membranes could be obtained by solution-cast.

^1^H-NMR spectroscopy was used to determine the macromolecular structure and the real DS of SPAES samples. As representative samples, Figure 2 shows the ^1^H-NMR spectra of SPAES_0.75 and SPAES_2.4; ^1^H-NMR spectra of SPAES_1.0, 1.3, 1.6, 1.9, and 2.1 are reported in Supporting Information (Appendix A). Signals relative to the presence of SHQ monomer and those relative to its interaction with BFPS give rise to five peaks: (e): doublet at 7.00 ppm; (h): wide doublet at 7.05 ppm; (i): overlapping with peak (c) at 7.15 ppm; (g): doublet at 7.45 ppm; and (f): doublet at 7.85 ppm. Due to the low amount of THB used, the presence of the branching agent cannot be detected from ^1^H NMR spectra. The presence of R-SO_3_^−^ groups along the polymer chains was quantitatively measured by applying Equation (1). The results obtained, listed in Table 2 (2nd column), along with the relative intensity of peak “g”, that increases as the DS gets higher, are in good agreement with the nominal DS and IEC data measured via PT analyses (Table 2, 3rd column), suggesting that the homogeneous synthesis was successfully performed without undesired degradation and side reactions. In addition, according to the progressive reduction of BFPS and BHP monomers content, as the DS increases *T*_g_ data trend reported in Table 2, 4th column show that glass transition temperatures have a tendency to become lower as the amount of SHQ monomer gets higher [30]. Despite this decrease, even at high SHQ content, high *T*_g_ values are obtained, with the lowest being 197.2 °C in the case of SPAES_2.4. SPAESs DSC curves are reported in Appendix A (in Appendix A).

The electrical conductivity of linear and branched SPAES pellets was investigated by IS analyses in the frequency range of (10^−1^ –10^7^) Hz at 25 °C and 1 V AC voltage. The results are shown in Figure 3. Even though, overall, SPEAS can be considered as an insulator due to its low electrical conductivity values even at high frequency, a correlation between the DS and the electrical conductivity can be observed at low frequencies. Only in the case of linear SPAES characterized by 0.75 meq R-SO_3_^−^
× g^−1^ of polymer a linear trend can be observed. Table 2 reports the electrical conductivity values recorded at 1 Hz. The lowest value of electrical conductivity is shown by linear SPEAS with 0.75 meq R-SO_3_^−^
× g^−1^ of polymer. By increasing the DS up to 1 meq R-SO_3_^−^
× g^−1^ of polymer, the value increases up to two orders of magnitudes (2.19 × 10^−11^ S × cm^−1^). The electrical conductivity of SPAES_1.3, having 0.1% of branching degree, slightly increases (5.95 × 10^−11^ S × cm^−1^) while for the others branched samples, characterized by a higher DS, the electrical conductivity increases by one order of magnitude for SPAES_1.6, SPAES_1.9, SPAES_2.1, and up to three orders of magnitude for SPAES_2,4 which is characterized by the highest values recorded (1.25 × 10^−8^ S × cm^−1^). In the case of the SPEAS_0.75 sample, a first-order capacitor assumption can be applied: as the frequency increases, the impedance of the sample decreases linearly [37]. By increasing the DS, the samples are characterized by a higher number of charge carriers which promotes the electrical conductivity. Nevertheless, a common linear trend is only present in the range of 104 to 107 Hz. For lower frequencies, all the samples deviate from linearity leading to ‘S’ shaped curve. These results indicate that at these frequencies, the chain rearrangement according to the electric field is typically stronger and is affected by the charge density. As the DS increases, better chains rearrangements are achieved, leading to a better charge mobility, hence higher electric conductivity values [37,38]. In this case, solution-like behavior is achieved. The linear trend of SPAES_0.75 can be attributed to a charge density not high enough to promote a successful chain rearrangement. However, chain rearrangement is not fast enough when the frequency is too high, so all the samples behave like simple capacitors at high frequencies. Passing from the lower value of SPAES_0.75 to the higher value of SPAES_2.4, the electrical conductivity changes up to five orders of magnitude, therefore these two samples were taken as references and used for further investigations.

### 3.2. PEDOT_SPAESs Synthesis and Characterization of: LSM, Electrical Conductivity, and SEM Analyses

PEDOT solutions doped with SPAES_0.75 (PEDOT:SPAES_0.75) and SPAES_2.4 (PEDOT:SPAES_2.4) were obtained by high concentration solvent-based emulsion polymerization of EDOT in the presence of SPAES (1% *w*/*w* with respect to EDOT) as doping agent, ferric sulphate as radical initiator and by varying the reaction solvent, i.e., DMF, DMAc, DMSO, and NMP. Here, all PEDOT:SPAES polymer solutions have excellent miscibility in each reaction solvent and appear brown in the case of DMF, orange for DMAc, green with DMSO, and purple in NMP (Figure 4), according to SPAES solvatochromic behavior [35]. On the other hand, in agreement with Sabatini et al. [34,35], undoped PEDOT is not soluble in the selected solvents and therefore it cannot be used as reference during the discussion of the data obtained.

PEDOT:SPAES_0.75 and PEDOT:SPAES_2.4 thin films were obtained by spin coating technique over SiO_2_ wafer and characterized in terms of thickness by LSM. Both films present a variable thickness: between 20 to 35 nm in the case of PEDOT:SPAES_0.75, and from 20 to 30 nm in the case of the PEDOT:SPAES_2.4. Confocal laser scanning microscope images and height profile of PEDOT:SPAES_0.75 and 2.4 thin films are shown in the Supporting Information, Appendix A in Appendix A.

Figure 5 shows a comparison of the electrical conductivity features of PEDOT:SPAES_0.75 (Figure 5a) and PEDOT:SPAES_2.4 (Figure 5b) prepared from DMF, DMAc, DMSO, and NMP as solvents. Resistance values were recorded by setting up a modified van der Pauw configuration, with the probes in contact along the thickness of the film. Electrical conductivity values are obtained by applying Equation (4), accounting for films’ geometrical factors (cross-sectional area and length) and assuming an average thickness of 25 nm for the films.

From Figure 5a,b, it is possible to observe that both solvent and DS have a high influence over the PEDOT: SPAES films electrical conductivity. According to the values of the electrical conductivity of SPAES (Table 2, 5th column), the content of sulfonic moieties seems to play a key role over the electrical properties of the resultant films: all samples with a DS of 2.4 meq R-SO_3_^−^
× g^−1^ of polymer have an electrical conductivity higher than SPAESs with lower DS.

On the other side, independently from the DS chosen, for all cases, the highest values of the electrical conductivity are achieved using DMSO as solvent. Furthermore, the values obtained with DMF, DMAc, and NMP overlap in the case of PEDOT: SPAES_0.75; for PEDOT: SPAES_2.4, even though the values are in a close range, better performances are obtained with DMAc and DMF, while the worst behavior is obtained using NMP. PEDOT_DMSO electrical conductivity passes from 65.03 (S × cm^−1^) when doped with SPAES_0.75 to 120.93 (S  ×  cm^−1^) with SPAES_2.4; in the case of PEDOT-based DMAc films the electrical conductivity goes from 37.85 to 77.84 (S  ×  cm^−1^) by increasing DS; lastly, PEDOT:DMF film shows electrical conductivity of 32.18 (S × cm^−1^) when doped with SPAES_0.75 and of 64.96 (S  ×  cm^−1^) with SPAES_2.4. This means that, by keeping constant film temperature at 373 K and passing from SPAES_0.75 to SPAES_2.4, the electrical conductivity doubles in the case of DMSO, DMAc, and DMF. The only solvent which does not follow the same trend is NMP. Indeed, at 373 K, the highest conductivity that can be achieved is around 37 (S  ×  cm^−1^) for both DS used. A typical increase of the electrical conductivity with increasing temperature can be observed which is consistent with the literature [37,38]. This trend is related to the increase in the mobility of the charge carriers derived from π-electrons, resulting overall in a higher charge density. The presence of a secondary dopant, such as the organic solvent in this case, encourages a rearrangement of the chain, according to which the charge carriers can pass more or less freely along the polymer backbone [8,39,40].

As it is shown in Table 2, the samples are all characterized by high *T*_g_ values and their DS have been determined by ^1^H-NMR and IEC. ^1^H-NMR appeared to be a robust tool to confirmed IEC. Knowing the real DS, it was possible to study its influence on the electric properties as discussed above and it resulted that the charge density influences chain rearrangement in the low frequencies field. Correlating the effect of DS and solvent on the electrical conductivity performances, the best result was obtained in the case of PEDOT: SPAES_2.4 sample solubilized in DMSO. To investigate its stability over time, PEDOT: SPAES_2.4 in DMSO film was tested in air through 15 heating-cooling cycles (from 373 to 313 K). The electrical conductivity values were recorded at 373 K and normalized by the highest value obtained, as reported in Figure 6.

During the first 5 heating-cooling cycles, the electrical conductivity of the films is not uniform. From cycle 6 up to cycle 9, it tends to an average constant value. The highest electrical conductivity is achieved in cycle 10. After that, the electrical conductivity shows only a slight decrease of about 5% with respect to the maximum value reached. This trend can be justified by considering the sensitivity of the film towards water provided by the presence of the R-SO_3_H hygroscopic groups. The film tends to adsorb water at each cooling step. The adsorbed water promotes little chain rearrangements, altering the charge movement along the polymer backbone and leading to higher resistivity of the film [41]. Water evaporates upon heating, hence a PEDOT:SPAES film became more concentrated in DMSO. Chains rearrangement is promoted and, at the same time, PEDOT partially tends to crystallize [42]. Over time, the film tends to absorb less water and to become more stable towards air exposure, resulting in a higher final electrical conductivity [42,43]. It is worth to note the mechanical stability of PEDOT: SPAES_2.4 film prepared starting by DMSO during the different heating-cooling cycles performed (15 heating-cooling cycles), i.e., the absence of cracking, delamination, and breaking phenomena that could affect the overall electrical conductivity performances of the materials studied.

To assess the solvent influence and the casting solvent method over PEDOT-based foil, the morphology of PEDOT: SPAES_2.4 films obtained in DMF, DMAc, DMSO, and NMP was evaluated via SEM. To better understand the influence of the solvent on the final surface morphology, PEDOT: SPAES_2.4 films were prepared by slow rotation technique, as previously described in the Materials and Methods section.

As widely reported in the literature [40,44,45], in the case of PEDOT films, four factors play a key role on the final morphology of the corresponding film and on the resulting conductive properties: solvent dipolar moment, dielectric constant, chemical structure, and the feed ratio between EDOT monomer and doping agent. About the role of the solvent, Kim et al. [40] showed that the presence of solvent traces on PEDOT doped with sulfonated polystyrene (PSS) films surface induces a screening effect between positively charged PEDOT chains and negatively charged PSS [44]. The higher the dielectric constant, the stronger resulted to be the screening effect. Furthermore, according to Shinar et al. [45], a solvent having two or more polar groups, such as DMF, DMSO, and NMP, is able to interact either with the dipole or the positive charge along PEDOT chains [45] and may induce chain rearrangements that result in the formation of a polymer foil with homogeneous morphology. An exception is represented by DMSO, which presents only one polar group. DMSO’s very high dipolar moment (4.1 D) and dielectric constant (46.68) favor mechanisms of solvent-polymer chains interactions and reorganization comparable to the ones previously described in the case of more polar solvents [40,45,46].

Regarding the influence of PEDOT and doping agent feed ratio onto film morphology properties, Crispin et al. [44] proposed that, during the synthesis of PEDOT-based polymeric dispersions, EDOT could be in the presence of PSS excess. Once the oxidizing agent is added, PEDOT oligomers formed result to be homogeneously spread in the solution [44], and PSS excess tends to surround PEDOT:PSS particles. When the dispersion is coated on the substrate, PSS excess aggregates in grains over the film surface. A high PSS grains excess dramatically affects PEDOT electrical conductivity due to the formation of a foil with irregular morphology and several defects (e.g., aggregates, clusters, holes, and so on); on the other side, a good balance between PEDOT: SPAES domain and dopant excess improves the conductive features of the final foil thanks to the formation of homogenous polymer surfaces.

SEM micrographs of PEDOT: SPAES_2.4 in DSMO film are shown in Figure 7. The film surface appears continuous, smooth (Figure 7a,b) and homogeneously covered by grains (Figure 7c,d), as in the case of PSS dopant excess previously discussed. Few cracks can be observed on the film surface as the consequence of solvent evaporation once the specimen was set in the high-vacuum SEM chamber. Getting closer to the film surface, cauliflower-shape grains [47,48] cover the surface of the polymer foil. The presence of grains on the film surface can be attributed to SPAES excess reorganization and aggregation process. Once the polymer dispersion is coated on the substrate, SPAES excess aggregates in grains over the film surface and high electrical conductivity values can be achieved, thanks to DMSO’s high dielectric constant that induces a good rearrangement and interaction between PEDOT and SPAES chains. Furthermore, according to Sotzing et al. [49], DMSO favors segregation phase phenomena between PEDOT and its dopant counterpart, in this case SPAES, improving electron hopping in PEDOT-rich areas. Lastly, a good solvent evaporation process ensures the formation of a close smooth film in which a homogenous grains distribution on the film surface can be observed.

The presence of SPAES grains is more pronounced in the case of DMAc as solvent (Figure 8). Figure 8a,b shows a highly inhomogeneous film surface. The continuous PEDOT: SPAES film surface is covered by SPAES grains which aggregate in clusters of different dimension and shape (Figure 8c,d). Considering the little variation in terms of DMSO and DMAc electrical dipole moment (4.1 D *vs* 3.72 D, respectively) [46], it could be possible that the DMAc extra polar group is able to induce a different PEDOT and SPAES chains rearrangement, which lead to a worst SPAES clean-off during the film formation process, and a lower electrical conductivity behavior in comparison to DSMO-based foil.

Figure 8c,d shows the result of SEM analyses carried out with PEDOT:SPAES_2.4 prepared from DMF. Two major domains can be observed, a continuous smooth domain attributed to PEDOT:SPAES blends and, on the wafer edges, a different domain characterized by holes and grains (Figure 8c,d). According to the results of the electrical conductivity analyses, even though PEDOT:SPAES_2.4 in DMF was following the same trend as DMSO, i.e., increasing electrical conductivity with the enhance of temperature, the final conductivity value reached by DMF was approximately half the one achieved with DMSO. This behavior can be explained considering that DMF presents two polar groups, thus it is able to engage strong interactions with PEDOT and SPAES. However, DMF dipole moment (3.86 D) and dielectric constant of 36.72 are lower than DMSO ones (4.1 D and 46.68, respectively), thus such kind of solvent is not able to induce a comparable chain rearrangement to DMSO [46].

Figure 8c,d, probably due to the fast solvent evaporation, leads to a discontinuity which affects also charge mobility and therefore conductivity performances. It has to be pointed out that, DMF, with a boiling temperature of 153.0 °C, is the more volatile solvent among the ones here used [46]. Lastly, the lower electrical conductivity, which characterizes DMF films, can be considered as deriving from a different rearrangement and interaction chains, which leads to the presence of big clusters (Figure 7d). In addition, it can be assumed that the clusters are present only on the edges because the polymeric dispersion prepared in DMF covers the substrate surface in a non-homogeneous way and migrates to the wafer edges, according to the centrifugal force applied during film formation process.

Figure 8e,f shows the SEM micrographs of PEDOT:SPAES_2.4 thin film prepared in NMP, which covers the entire wafer surface. The film is inhomogeneous but did not show any SPAES aggregates. NMP presents two polar groups as DMF and DMAc, and a dipole moment of 4.09 D comparable to DMSO. Only its dielectric constant (32.02) is lower than the ones reported for the other solvents, but is quite close to DMF and DMAc data [46]. Thus, its worst electrical conductivity can be attributed not to the nature of the solvent itself, but to its capability of film-forming with PEDOT:SPAES polymers. It can be assumed that in case of NMP, the final PEDOT:SPAES polymer solution tends not to cover homogeneously the substrate surface, which probably is due to the poor affinity between solvent and polymers. The absence of grains affects conductivity data too and is a further evidence of NMP’s low capability to promote interaction and rearrangement of PEDOT and SPAES chains. Even though a SPAES excess, high or low, affects the electrical conductivity as in the case of DMAc/DMF, a good balance between PEDOT:SPAES domain and SPAES excess, as in the case of DMSO, optimizes chains interlink. Lastly, according to the previously discussed NMP-based samples electrical conductivity data, i.e., the no correlation of SPAES DS onto PEDOT film electrical conductivity properties, these results confirm the straight relationship between film morphology and performances of the same polymer foil.

## 4. Conclusions

In the field of conductive polymers, the development of poly(3,4-ethylenedioxythiophene) (PEDOT)-based materials characterized by enhanced electrical conductivity and solubility in common inorganic and organic solvents is always a pressing request. In this context, sulfonated polyarylether sulfone (SPAES) used as PEDOT doping agent can be a way to create tailor-made materials with enhanced electrical conductivity.

Here, PEDOT_SPAES were successfully synthesized via oxidative polymerization using SPAES with increasing DS as dopant. As described, SPAES homogenous synthesis provides multiple advantages, including straight control over the final DS and water content; in fact, SPAES R-SO_3_H groups act as a template in the formulation, stabilizing the positive charge along PEDOT backbone and improving the final hydrophilicity of the material, leading to PEDOT_SPAES solutions with enhanced solubility in common organic solvents such as *N*,*N*-dimethylformamide (DMF), dimethylacetamide (DMAc), dimethyl sulfoxide (DMSO), and *N*-methyl-2-pyrrolidone (NMP).

Electrical conductivity in SPAES samples was studied via impedance spectroscopy studies, showing that the high content of sulfonic groups promotes PEDOT’s conductivity. Here, for the first time, a correlation between the DS of SPAES and the electrical conductivity of the resulting PEDOT_SPAES thin films obtained comparing different polymerization solvents (DMF, DMAc, DMSO, and NMP) was established. PEDOT_SPAES with a DS of 2.4 meq R-SO_3_^−^
× g^−1^ of polymer solubilized in DMSO appeared to be the best candidate, reaching an electrical conductivity of 120 (S  ×  cm^−1^) at 373 K, which is stable for at least 15 heating/cooling cycles.

Morphological studies of PEDOT_SPAES_2.4 films obtained from the four solvents previously cited were performed via SEM analyses. Here, DMSO, DMAc, and DMF-based films present a continuous domain attributed to PEDOT:SPAES blends onto which a second domain, SPAES excess, is distributed. However, a too high concentration of dopant aggregates, as in case of DMAc-based foil, a too low presence of doping agent grains as for DMF-based sample, or the absence of doping agent aggregates as in the case of NMP foil, results in lower electrical conductivity compared to the DMSO-based sample, where doping agent aggregates cover homogenously the surface film. Correlating doping agent sulfonic moieties content and DMSO’s capability to form PEDOT:SPAES foils with a smooth surface homogeneously covered by SPAES grains, the best performances for PEDOT-based foils in terms of electrical conductivity can be achieved using SPAES with high DS and DMSO as a casting solvent.

## Figures and Tables

**Figure 1 polymers-13-00658-f001:**
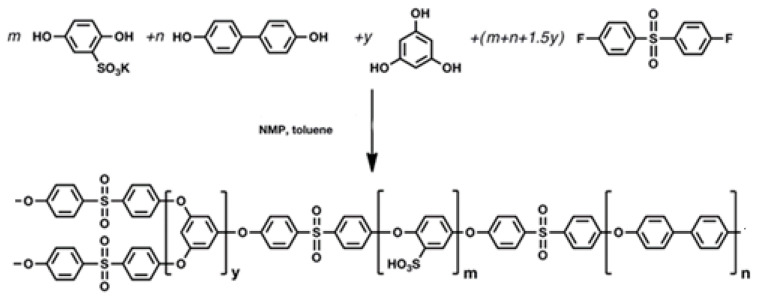
Scheme of the one-pot polymerization of branched SPAESs, where A stands for the aryl halide reactive group, B for the aryl hydroxy group, and y for THB concentration. Stoichiometric coefficients are indicated with *m* for SHQ, *n* for DHDP, *y* for THB, and (m + n + 1.5y) for BFPS.

**Figure 2 polymers-13-00658-f002:**
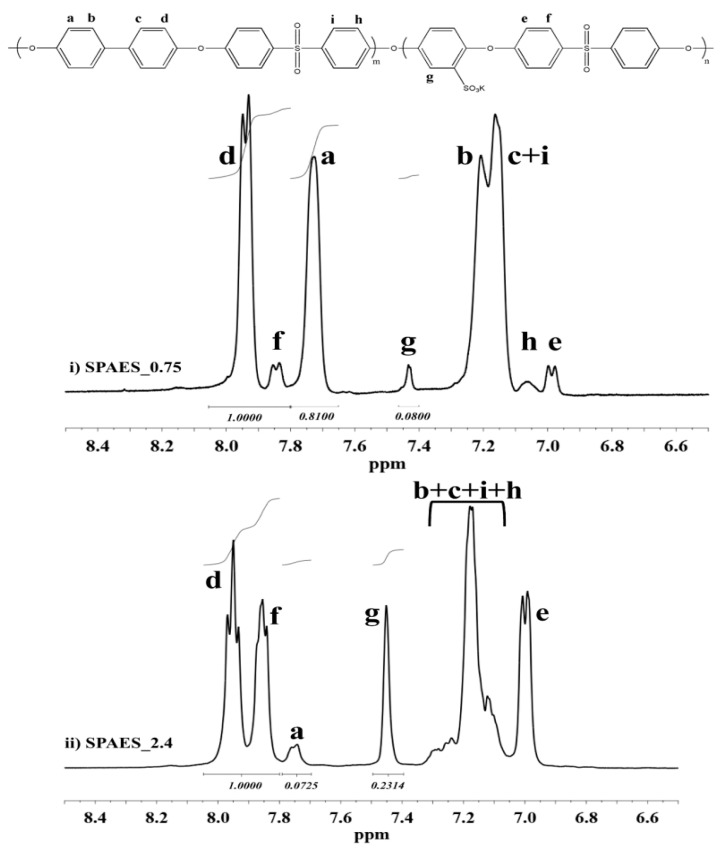
^1^H-NMR spectra aromatic region of (**i**) linear SPAES_0.75 and (**ii**) branched SPAES_2.4.

**Figure 3 polymers-13-00658-f003:**
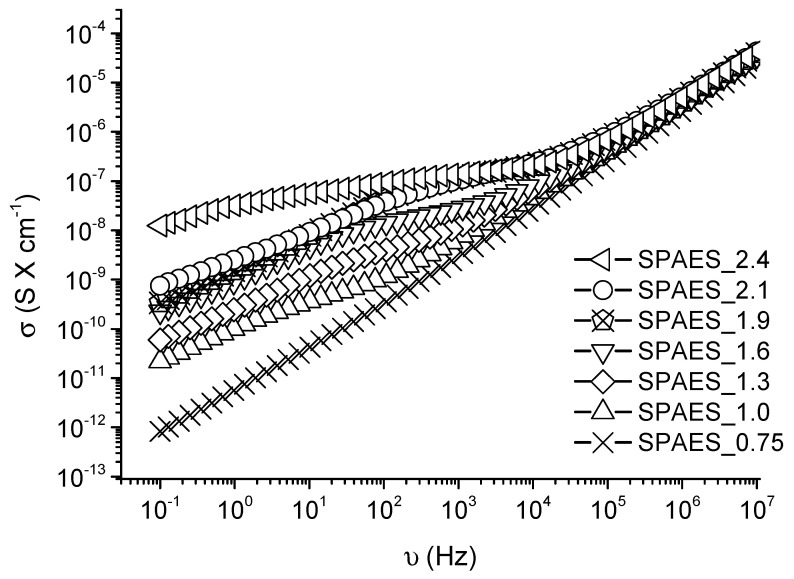
Electrical conductivity σ against frequency υ of SPAES samples: (
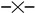
) DS = 0.75, (
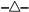
) DS = 1.0, (
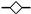
) DS = 1.3, (
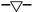
) DS = 1.6, (
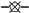
) DS = 1.9, (
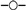
) DS = 2.1, and (
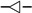
) DS = 2.4 (meq R-SO_3_^−^
× g^−1^ of polymer) measured via impedance spectroscopy. An increasing trend of electrical conductivity with increasing DS can be observed, especially at low frequencies.

**Figure 4 polymers-13-00658-f004:**
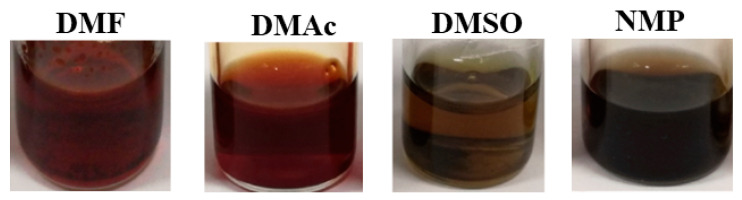
Solvatochromic effect of SPAES with DS = (2.4 meq R-SO_3_^−^
× g^−1^ polymer) when used as doping agent of PEDOT in DMF, DMAc, DMSO, and NMP solvents.

**Figure 5 polymers-13-00658-f005:**
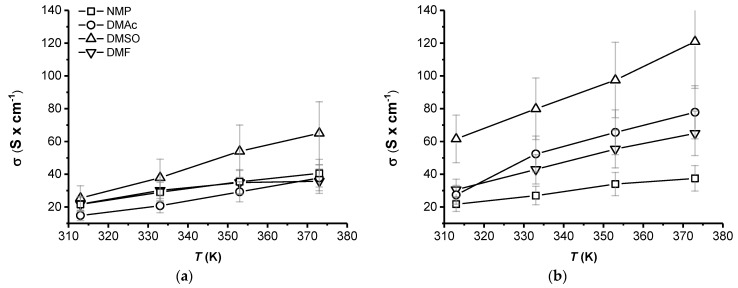
Electrical conductivity *σ* (S × cm^−1^) values against temperature *T* (K) of PEDOT films doped with SPAES having a DS of (**a**) 0.75 and (**b**) 2.4 (meq R-SO_3_^−^
× g^−1^ polymer) and dissolved in (-∇-) DMF, (-○-) DMAc, (-Δ-) DMSO, and (-□-) NMP. The films were prepared by spin coating technique. Resistance values were recorded with a multimeter in DC regime.

**Figure 6 polymers-13-00658-f006:**
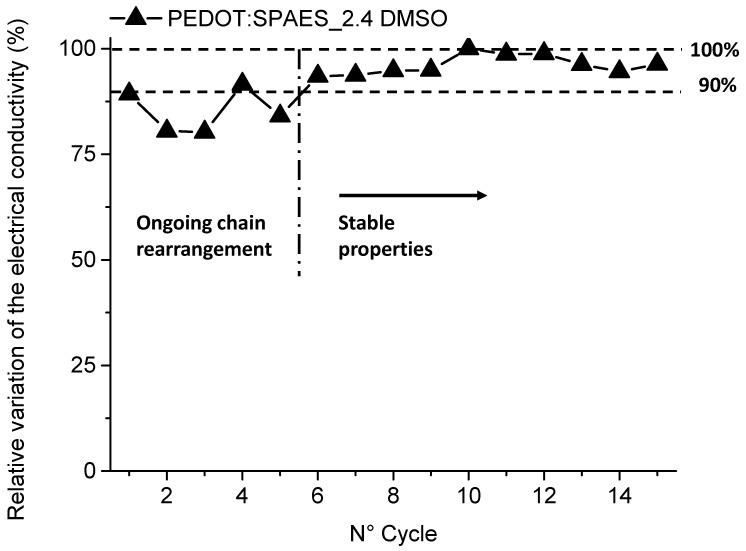
Relative electrical conductivity of PEDOT: SPAES_2.4 in DMSO after having exposed the film to 15 heating-cooling cycles (from 298 to 373 K). The values shown are the measured values at 373 K at the beginning of each cycle and are normalized by the highest value obtained.

**Figure 7 polymers-13-00658-f007:**
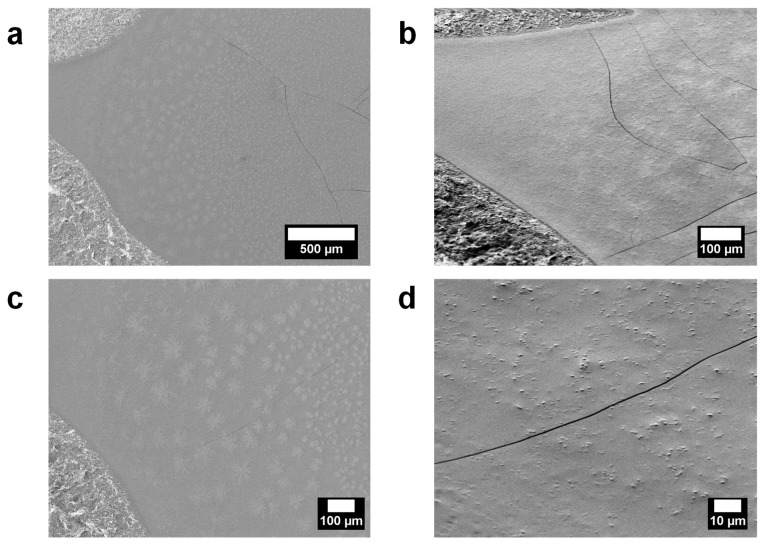
SEM micrographs of PEDOT:SPAES_2.4 thin film in DMSO with silver paste on the SiO_2_ wafer edges as contrast. The film was prepared by slow rotating 15 µL of polymeric solution for 24 h at room temperature. The top view of the specimen is shown in micrographs (**a**,**c**) where it is possible to observe a close inhomogeneous film coated with SPEAS clusters. On the surface, there are few cracks that formed once the specimen was placed under the SEM high vacuum chamber. SEM micrographs of the tilted specimen are shown in (**b**,**d**), in which different shapes of SPEAS clusters are reported.

**Figure 8 polymers-13-00658-f008:**
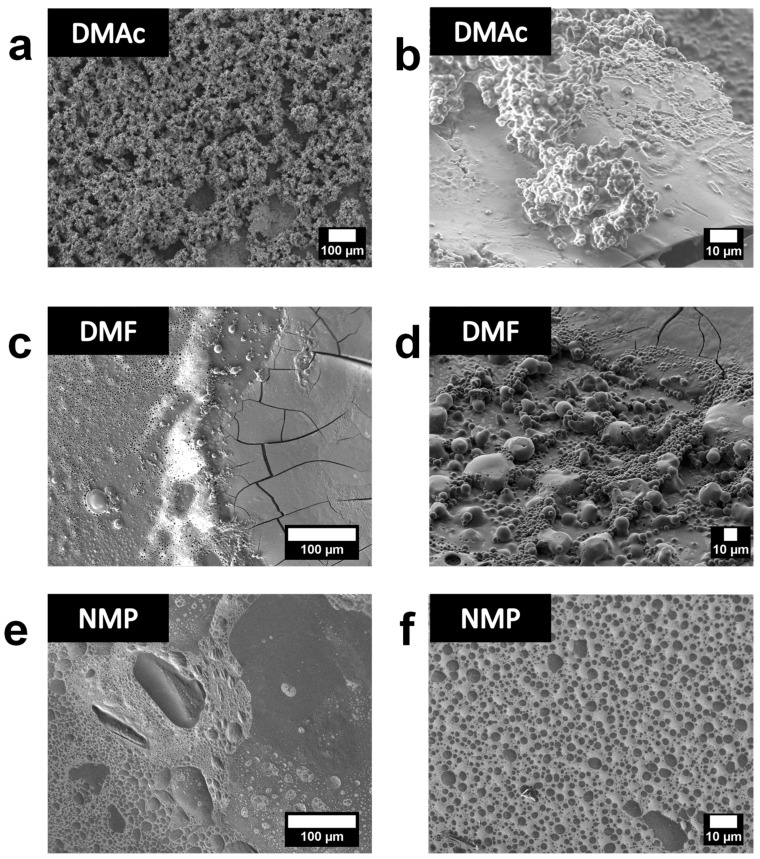
SEM micrographs of PEDOT:SPAES_2.4 thin films in the chosen solvents: (**a**,**b**) DMAc; (**c**,**d**) DMF; and (**e**,**f**) NMP.

**Table 1 polymers-13-00658-t001:** Amounts of reagents used for the synthesis of sulfonated polyarylether sulfone (SPAES) samples.

Sample	Nominal DS (meq R-SO_3_^−^ × g^−1^ of Polymer)	BFPS (g)	BHP (g)	SHQ (g)	THB (g)	K_2_CO_3_ (g)
SPAES_0.75	0.75	3.074	1.553	0.856	-	3.677
SPAES_1.0	1.0	3.041	1.296	1.141	-	3.637
SPAES_1.3	1.3	3.002	0.982	1.484	0.003	3.591
SPAES_1.6	1.6	2.962	0.673	1.826	0.003	3.543
SPAES_1.9	1.9	2.922	0.365	2.168	0.003	3.495
SPAES_2.1	2.1	2.896	0.159	2.398	0.003	3.463
SPAES_2.4	2.4	2.876	0.005	2.569	0.003	3.439

**Table 2 polymers-13-00658-t002:** Results of SPAESs characterization in terms of real DS measured via ^1^H NMR spectroscopy, IEC determined via PT analyses, *T*_g_ evaluated by means DSC measurements and values of the electrical conductivity estimated via impedance spectroscopy at a frequency of 1 Hz.

Sample	Real DS (meq R-SO_3_ × g^−1^ of Polymer) via ^1^H NMR spectroscopy	IEC (meq R-SO_3_ ^*^ g^−1^ of polymer) via PT Analyses	T_g_ (°C)	Electrical Conductivity (S × cm^−1^)
SPAES_0.75	0.70	0.74	290.9	8.26 × 10^−13^
SPAES_1.0	0.94	0.90	295.3	2.19 × 10^−11^
SPAE_1.3	1.33	1.30	245.3	5.95 × 10^−11^
SPAES_1.6	1.52	1.55	243.5	2.23 × 10^−10^
SPAES_1.9	1.83	1.87	~241.3	3.38 × 10^−10^
SPAES_2.1	2.00	2.05	~230.4	7.49 × 10^−10^
SPAES_2.4	2.20	2.25	197.2	1.25 × 10^−8^

## Data Availability

The data presented in this study are available on request from the corresponding author.

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
