# Peer review of "Role of Doping Agent Degree of Sulfonation and Casting Solvent on the Electrical Conductivity and Morphology of PEDOT:SPAES Thin Films"

_polymers, 2021, doi:10.3390/polym13040658_

Round 1
Reviewer 1 Report
This work mainly discusses the effect of the degree of sulfonation (DS) in SPAESs on the electrical conductivity of PEDOT: SPAES in different solvents. A higher DS and a smoother thin film were determined to be beneficial for the conductivity.
- Page 7, line 206: It is not clear why the author synthesized linear SPAES and branched SPAES with different DS. How do the branched SPAES differ from liner SPAES? Please also include the chemical structure for branched SPAES.
- Table 2 provides the Tg values. However, both TGA and DSC results for all SPAES are missing here. Please provide it in the supporting information. Reference 16 only shows one set of DSC data. Meanwhile, it is unclear why a Tg shows up at > 200 °C while TGA shows degradation starting from 100 °
- Page 9, line 235: Here, the sample linear SPAES_0.75 shows a near-linear improvement of conductivity with frequency from 0.1 to 107 All branched SPAES show an “S” shaped curve with a slower increase from 10 to 104 Hz and a faster increase from 104 to 107 Hz. Please specify the origin of such slope differences between different samples.
- It is suggested to discuss the advantages of SPAES over PSS in terms of the effect on the conductivity.
- PEDOT: PSS is commonly processed in a water solution. Can the authors comment on why the water was not used as the solvent here?
- Figure 4: Please include error bars in the figure.
- Page 9, line 236: “con” is misspelled.
Reviewer 2 Report
The manuscript by Tomasino et al. presents an very interesting study on the development of SPAES as doping agent for PEDOT with enhanced electrical conductivity (w DMSO as secondary solvent). The authors used various characterization techniques and investigated the role of doping agent degree of sulfonation and casting solvent on the conductivity and morphology of PEDOT:SPAES film. The results shed light on potential applications as electronic materials. The experimental results and interpretations are sound. And thus I recommend publishing this manuscript after minor revision. The specific comments are as follows:
1) What is the advantage/disadvantage of PEDOT:SPAES compared with PEDOT:PSS? I suggest authors to include brief discussion on this part.
2) I recommend authors to add error bars for the data points in Figure 4 and 5.
3) For introduction section, I suggest authors to introduce PEDOT as conductive polymer first (bring Page 3, line 69 to the beginning). And then introduce SPAES as ionomer dopant as replacement to the conventional PSS dopant. Basically to reverse the paragraphs of SPAES and PEDOT. This will create a better flow in the introduction part.
4) One of the merits of a conductive polymer is to obtain the flexibility of the device (ie. flexible electronics). It should be good idea if the merits are addressed in the introduction section.
5) One of the benefit of DMSO is it helps create phase segregation between PEDOT and its dopant counterpart. This contribute to an improved electron hopping in PEDOT-rich regions. I suggest authors to include brief discussion on this part. I have include a paper below you can reference on.
Ref: Otley M, Alhashmi Alamer F, Guo Y, Santana J, Eren E, Li M, Lombardi J, Sotzing G (2016) Phase segregation of PEDOT:PSS on textile to produce materials of > 10 A mm−2 current carrying capacity. Macromol Mater Eng 302:1600348
6) Please comment the mechanical property/stability of the PEDOT:SPAES film.
Reviewer 3 Report
The contribution is clearly defined in the summary. Identifying the novelty and importance of the study.
The state of the art is adequate and details the various novelties and characteristics of the materials under study. It is described and supported with appropriate literature.
The methodology is clear, detailing the methodology of research
Letters below spectrum (Figure 1), may be clearer. Improve quality.
The discussion associated with the results shown in Table 2 can be improved.
Additional information is very useful and is adequately included
Document displays apparent corrections in red coloration, highlighting support information.
The discussion of results is supported by scientific literature, which allows to support the conclusions indicated.
Round 2
Reviewer 1 Report
All the comments have been well-addressed.
